# Wastewater Management Using Coagulation and Surface Adsorption through Different Polyferrics in the Presence of TiO₂-g-PMAA Particles

Heba Saed Kariem Alawamleh [1], Seyedsahand Mousavi [2], Danial Ashoori [3], Hayder Mahmood Salman [4], Sasan Zahmatkesh [5,†] and Mika Sillanpää [6,7,*,‡,§]

1 Department of Basic Scientific Sciences, Al-Huson College, AL-Balqa Applied University, P.O. Box 50, Al-Huson 21510, Jordan
2 Department of Environmental Engineering, Faculty of Civil, Water & Environmental Engineering, Shahid Beheshti University, Tehran 1983969411, Iran
3 Department of Apply Chemistry, Faculty of Chemistry Science, Islamic Azad University Tehran-North Branch, Tehran 1983969411, Iran
4 Department of Computer Science, Al-Turath University College Al Mansour, Baghdad 10070, Iraq
5 Tecnologico de Monterrey, Escuela de Ingenieríay Ciencias, Puebla 72453, Mexico
6 Faculty of Science and Technology, School of Applied Physics, University Kebangsaan Malaysia, Bangi 43600, Malaysia
7 International Research Centre of Nanotechnology for Himalayan Sustainability (IRCNHS), Shoolini University, Solan 173212, India
* Correspondence: mikaesillanpaa@gmail.com
† Current Adress: Department of Chemical Engineering, University of Science and Technology of Mazandaran, Behshahr P.O. Box 48518-78195, Iran.
‡ Current Address: Department of Chemical Engineering, School of Mining, Metallurgy and Chemical Engineering, University of Johannesburg, P.O. Box 17011, Doornfontein 2028, South Africa.
§ Current Address: Department of Civil Engineering, University Centre for Research & Development, Chandigarh University, Gharuan, Mohali 140413, India.

**Abstract:** For the surface adsorption process, a wide range of studies have been carried out to describe the adsorption process. However, no extensive study has been carried out to investigate the pre-treatment method effect on the separation process. The purpose of the present study is to improve the performance of the membrane process in the treatment of oily wastewater. For this purpose, the effects of pre-treatment, membrane modification, and operational parameters were investigated on the microfiltration membrane system. Two methods of coagulation and surface adsorption were used as pre-treatment, and then a modified polysulfone (PSf) membrane containing TiO₂ nanoparticles was applied in the microfiltration process. In order to reduce the membrane fouling and increase the permeate flux, the surface of the nanoparticle was modified. In order to check the performance of coagulation, pretreatments of polyferric sulfate (PFS) and polyferric chloride (PFC) were applied. The results showed that the Chemical Oxygen Demand (COD) reduction of 98% can be obtained using 1 g/L of PFS coagulant at pH = 6, while only 81% of COD was removed using 1 g/L PFC. It was also found that the best pH for the performance of this type of coagulant was measured as pH = 10 and the removal efficiency for 1 and 2 g/L of PFC coagulant was obtained as 96.1% and 91.7%, respectively. The results show that in the case of using a coagulant of less than 1 g/L, using PFS is more efficient than PFC; meanwhile, in more than 1 g/L of coagulant, this effect is reversed and the use of PFS will be less efficient than PFC. The performance of the PSf-TiO₂ membrane fabricated by the Nonsolvent-induced phase separation (NIPS) method was investigated using modified nanoparticles with an initial size of 10 nm at different operating conditions. The results show that the permeate flux and the rejection can be increased to 563 L/h m² and 99%, respectively, using the modified PSf membrane. The results of this paper showed that the performance of the adsorption process can be improved by using the coagulation process as a pre-treatment method.

**Keywords:** membrane separation; pre-treatment; coagulation; surface adsorption; oily wastewater

## 1. Introduction

The impressive progress of various industries and the increase in the amount of produced wastewater, as well as the challenging issue of water scarcity, have made the wastewater treatment a vital and inevitable matter [1–3].

Oily wastewaters are considered as one of the most important sources of pollution, which are obtained from many industries such as oil [2] and gas, metalworking, as well as food industries [4]. So far, various methods have been studied and investigated in the field of oily wastewater treatment, such as flotation [5], separation based on gravity [6], surface adsorption, coagulation [7] and flocculation [8], and ozonation [9], etc. Nevertheless, most of these methods have not been satisfactorily applied in terms of efficiency in the treatment of oily emulsions with an oil particle size of less than 20 μm [10].

Recently, membranes were reported as an attractive method for oily wastewater treatment [9] due to their low cost and energy consumption [11,12]. However, these methods have disadvantages such as fouling and concentration polarization, which can reduce the life of the membrane [13]. Therefore, in order to increase the membrane efficiency, it is possible to act in two ways: (a) preventing the particles in the effluent from reaching the surface of the membrane and (b) surface washing of the membrane [14].

In order to reduce membrane fouling, many methods were previously investigated, among these methods [10], pre-treatment of the wastewater and modification of the hydrophilic property of the membrane are the most important methods, which leads to an increase in the number of pores on the surface and an increase in the flux [15]. Among the common methods used in pre-treatment, the combination of two physical and chemical methods, surface adsorption [11] and chemical coagulation, was given much attention due to its low cost and high performance [16].

In surface adsorption, common adsorbents such as activated carbon and zeolites have provided a strong attraction to a wide range of dissolved particles [15], including colloids, due to their specific surface area and high surface activity [17,18]. On the other hand, coagulants such as iron sulfate, iron chloride, aluminum sulfate, and Polyferric chloride (PFC) are widely used in pre-treatment by chemical coagulation methods due to their availability and low prices [19].

According to the previously reported studies, the type and amount of coagulant [15] and the properties of coagulated particles [7] such as their surface charge in the removal of organic particles in wastewater are one of the most important parameters affecting the process [20,21]. As a result, reducing the accumulation of solid particles dissolved or suspended on the surface of the membrane or modifying the formed coagulates, which leads to the reduction of the phenomenon of membrane fouling, are methods that can obviously affect the treatment process [19]. Another method that has had a significant effect in improving the membrane performance and reducing the membrane fouling is increasing the hydrophilic property of the membrane surface, which has received a lot of attention in recent years [22,23].

Theoretically, membranes can be produced from all of the polymers, but only a few polymers are practically used in membrane production [24], which depends on its physical and chemical properties. Polysulfone (PSf) is a thermoplastic polymer that is widely used in the manufacture of polymer membranes [12] due to its favorable characteristics such as good chemical and thermal stability, high mechanical strength, and high glass transition temperature [25].

In addition to the advantages mentioned for Polysulfone (PSf), it should be kept in mind that this material is hydrophobic enough [22], and as a result, the water permeability has not been satisfactory for practical applications. Recently, some researchers have presented various methods to modify the PSf membranes and improve their properties, for example, by combining hydrophilic and hydrophobic polymers, and grafting hydrophilic branches on the polymer [26].

In addition to combining organic polymers, substantial research has been carried out on the combination of organic and inorganic materials and using nanoparticles in polymer

solutions, which leads to improving membrane performance and controlling its surface properties [27,28].

Various types of nanoparticles such as $TiO_2$, $Al_2O_3$, $ZrO_2$, $SiO_2$, and $Fe_2O_3$ are used to modify the polymer membranes [29–31]. Among the mentioned nanoparticles, $TiO_2$ is one of the most widely used because it has favorable chemical properties and suitable performance in the field of polymer membrane modification [32]. In order to reduce the clumping phenomenon of nanoparticles and increase the scattering properties, the surface of nanoparticles is modified by creating a hydrophilic polymer chain on the surface of the membrane [33]. In 2021, Tomczak and Gryta [34] investigated the microfiltration (MF) process of oily wastewaters using capillary polypropylene (PP) membranes. Furthermore, they studied the applicability of the ultrafiltration ceramic membrane for the separation of oily wastewaters generated during maritime transportation and found that the pre-filtration of oily wastewaters adversely affects the permeate flux [35].

It should be noted that the efficiency of a cross-flow membrane process depends on operational parameters such as cross-flow speed, operating pressure, temperature, membrane pore size, and the concentration of suspended particles in the feed [36].

For the surface adsorption process, a wide range of studies have been carried out to describe the adsorption process. However, no extensive study has been carried out to investigate the pre-treatment method effect on the surface adsorption process. Therefore, the coagulation and surface adsorption methods have not been fully understood. In this paper, in order to study the surface adsorption process, different pre-treatment methods have been performed for the removal of oily pollutants. In this regard, the obtained experimental data have been carefully discussed to suggest the optimum method. The aim of this paper is to use two coagulation and surface adsorption methods using polyferric chloride (PFC) and polyferric sulfate (PFS) coagulants and activated carbon adsorbent as pre-treatment methods. In this way, the modified $TiO_2$-g-PMAA membrane was prepared and the membrane performance was investigated by changing the operational parameters. The performance of the membranes was studied based on the permeate flux measurement and the measurement of the oil content in the permeate.

## 2. Materials and Methods

Polysulfone (PSf) with a molecular weight of 68,000 gr/mol was purchased from BASF. Polyvinyl pyrrolidone (PVP) with a molecular weight of 30,000 gr/mol which is applied as an additive and 1-methyl-2-pyrrolidone (NMP) were obtained from Merck. Double distilled water was used as a non-solvent in the coagulation bath. The $TiO_2$ nanoparticles with particle size of 10 nm have been used to improve membrane properties, and in addition, Tri AminoPropylTriMethoxySilane (APTMS) has been used to activate the nanoparticle surface and prepare it for polymerization. In order to prepare an emulsion with high stability in the feed, Tween 80 has been used as an emulsifier. In order to adjust the samples pH, sodium hydroxide, and 0.1 N hydrochloric acid were used. The polyferric chloride (PFC) powder and Polyferric sulfate (PFS) solution were used as coagulants. Activated carbon with the specifications mentioned in Table 1 was applied as an adsorbent.

**Table 1.** The specifications and characterization of the obtained activated carbon.

| Parameter | Specification |
|---|---|
| Iodine Number (mg/g) | 1093 mg/g |
| Hardness | 88% |
| Density (kg/m$^3$) | 422 kg/m$^3$ |
| Ash (%) | 2.4% |
| Humidity (%) | <9% |
| pH | 7 |
| Specific Surface (m$^2$/g) | 1110 m$^2$/g |

In order to check the performance of coagulation and to determine the optimal conditions in the application of the coagulation method, the jar test (FP4, Velp scientifica, Usmate, Italy) is applied, which includes 4 stainless steel containers of 1 L and two types of PFC coagulants due to the scope of operation [37]. The jar test operating conditions in all experiments were kept constant according to the following procedure: 60 s of rapid mixing at 140 rpm, 20 min of slow mixing at 40 rpm, and one-hour of settling time.

In order to obtain the equilibrium time, the samples with a concentration of 3000 ppm were prepared. In each of the samples, a specific amount (4 g/L) of activated carbon granules was poured and placed on a shaker at a speed of 200 rpm. Then, 10 different samples were taken in 5 h in 30-min intervals. After the analysis, the contact time was obtained. It should be noted that in all stages the temperature is adjusted to the ambient temperature [38].

In order to investigate the effect of pH value on oil adsorption, samples were prepared from a solution with an initial concentration of 3000 ppm, and the pH of these samples was adjusted by adding a solution of 0.1 N hydrochloric acid and 0.1 N sodium hydroxide. It should be noted that the pH adjustment was carried out using a sample dropper and the pH of the solutions is controlled by a digital pH meter (Metrohm, model 750, Riverview, FL, USA). After pH adjustment, the samples were transferred into a shaker to reach the equilibrium. In order to achieve the optimal amount of adsorbent in oil removal, different amounts of adsorbent were used in water-oil emulsion samples with an initial concentration of 3000 ppm [39].

In this paper, to measure Chemical Oxygen Demand (COD), the open reflux method (5220A-D) was applied using a spectrophotometer (JASCO, model V-550, Japan), which has less error than the titration method. This analysis has been carried out using the common potassium-dichromate oxidation method and using the EPA 410.4 standard [40]. In this method, by using the adsorbance reading from the spectrophotometer for the samples made according to the standard, the COD value is extracted by the calibration chart and finally, the reduction rate is calculated from Equation (1), which, $C_f$ and $C_p$ are the feed and permeate concentration, respectively.

$$R(\%) = \frac{c_f - c_p}{c_f} \times 100 \tag{1}$$

In order to modify the TiO$_2$ nanoparticles, 2.5 g of completely dried nanoparticles were firstly poured into 100 mL of toluene and ultrasonication was applied for 45 min to disperse the nanoparticles in the solvent. Then, 13 mL of APTMS as an activating agent is added to the solution. The reaction is carried out for 24 h at 97 °C. Finally, the solution is centrifuged, and the obtained product is washed three times with deionized water. The obtained activated TiO$_2$ nanoparticles are completely dried in an oven at 95 °C for 24 h [41].

In order to fabricate the membranes, first of all, the Polysulfone (PSf) is placed under vacuum in the oven for 12 h at 110 °C to completely dry. The process of preparing the solution is that first nanoparticles are added to the solvent and an ultrasonic device is used (for 45 min) to distribute nanoparticles uniformly. In this way, to prevent evaporation of the solvent, the solution container is placed in an ice bath and then sonication is performed. Finally, the PSf polymer and polyvinylpyrrolidone are added to the uniform solution in the desired amount [42].

In order to prevent the polymer from clumping, this substance is added to the solution over a long period of time, about an hour. Then the solution is stirred for 24 h at 1000 rpm using a stirrer. After complete dissolution, the solution is kept still for 12 h to remove the bubbles that were created during stirring as the presence of bubbles in the solution causes problems in the process of membrane fabrication.

To examine the structure and observe the surface and cross-sectional surface of the membrane, a Scanning Electron Microscope (SEM) was used with an accuracy of 5 nm and a magnification of up to 30,000. In this way, the prepared samples are, first broken in liquid

nitrogen, because otherwise, breaking the samples will change the structure due to the applied force.

## 3. Results and Discussion

The average oil particle size in the synthetic wastewater is reported as 105.6 nm as seen in Figure 1.

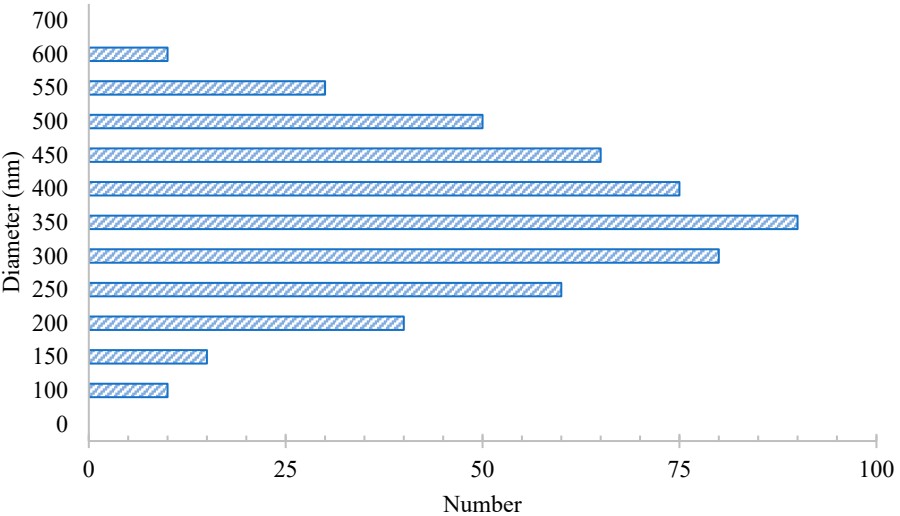

**Figure 1.** The average size of oil particles in the emulsion.

As mentioned, the pH value is an effective factor not only on the surface charge of coagulants, but also on emulsion stability, and it is the most important parameter affecting the coagulation process. In this way, a suitable pH value, in addition to neutralizing the colloidal particles with a negative charge and connecting these particles together, also helps in the formation of coagulates and their settling process [43].

For this purpose, the effect of pH value was investigated in fixed amounts of coagulant (1 and 2 g/L) and the results are given in Table 2. It should be noted that the initial COD of synthetic effluent was measured as 7500 ppm. Moreover, the pH value range was selected based on optimum performance of previous studies. For a better comparison, the rejection values in different PFS concentrations are shown in Figure 2.

**Table 2.** The pH effect on COD removal and the rejection using different amount of PFS as a coagulant.

| PFS | pH = 4 | | pH = 5 | | pH = 6 | | pH = 7 | | pH = 8 | | pH = 9 | | pH = 10 | |
|---|---|---|---|---|---|---|---|---|---|---|---|---|---|---|
| | COD | %R | COD | %R | COD | %R | COD | %R | COD | %R | COD | %R | COD | %R |
| 1 g/L | 412 | 97.1 | 394 | 98.2 | 381 | 98.3 | 591 | 91.3 | 673 | 80.8 | 1520 | 68.3 | 2913 | 55.0 |
| 2 g/L | 533 | 93.3 | 475 | 95.2 | 411 | 96.5 | 669 | 88.7 | 892 | 73.2 | 1737 | 56.8 | 3166 | 48.3 |

As can be seen in Table 2, the pH value is an effective parameter that can change the performance of polyferric sulfate coagulant. The highest removal efficiency in both cases (1 and 2 g/L of PFS), is evaluated at pH = 6. The results show that upper pH values decrease the efficiency, that can be caused by the formation of negatively charged ions such as $Fe(OH)_4^-$ which leads to the creation of electrostatic repulsion between ions and organic particles with a negative charge in the wastewater and prevents the formation of larger coagulates [44].

Therefore, in order to maintain the coagulation function in the alkaline pH range, it is necessary to consume more polyferric sulfate coagulant. It should be noted that the removal rate at optimal pH 6 for 1 and 2 g/L of PFS is 98.3% and 96.5%, respectively.

Table 3 shows the pH effect on COD removal and the rejection using different amounts of PFC as a coagulant. The results show that the highest removal efficiency in both cases

(1 and 2 g/L of PFS) can be obtained at pH = 10. In order to have a better comparison, the rejection values in different PFS concentrations are shown in Figure 3.

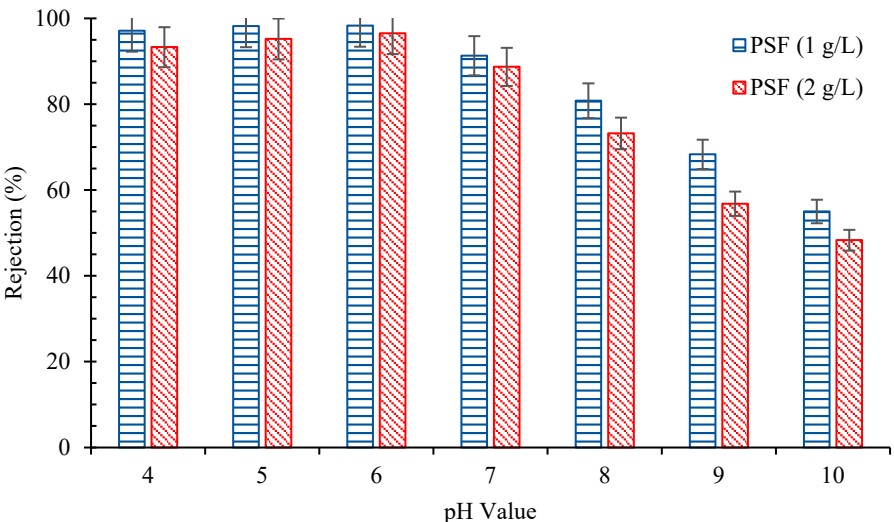

**Figure 2.** The comparison of the rejection values at different pH values using different concentrations of PFS. The upper and lower 95% confidence intervals are shown as error bars that extend above and below the top of the mean column.

**Table 3.** The pH effect on COD removal and the rejection using different amount of PFC as a coagulant.

| PFC | pH = 4 | | pH = 5 | | pH = 6 | | pH = 7 | | pH = 8 | | pH = 9 | | pH = 10 | |
|---|---|---|---|---|---|---|---|---|---|---|---|---|---|---|
| | COD | %R | COD | %R | COD | %R | COD | %R | COD | %R | COD | %R | COD | %R |
| 1 g/L | 1559 | 71.3 | 1174 | 75.9 | 948 | 81.3 | 766 | 84.7 | 512 | 88.3 | 319 | 93.3 | 188 | 96.1 |
| 2 g/L | 1945 | 65.2 | 1463 | 69.1 | 1248 | 73.9 | 988 | 79.8 | 716 | 83.2 | 533 | 88.2 | 397 | 91.7 |

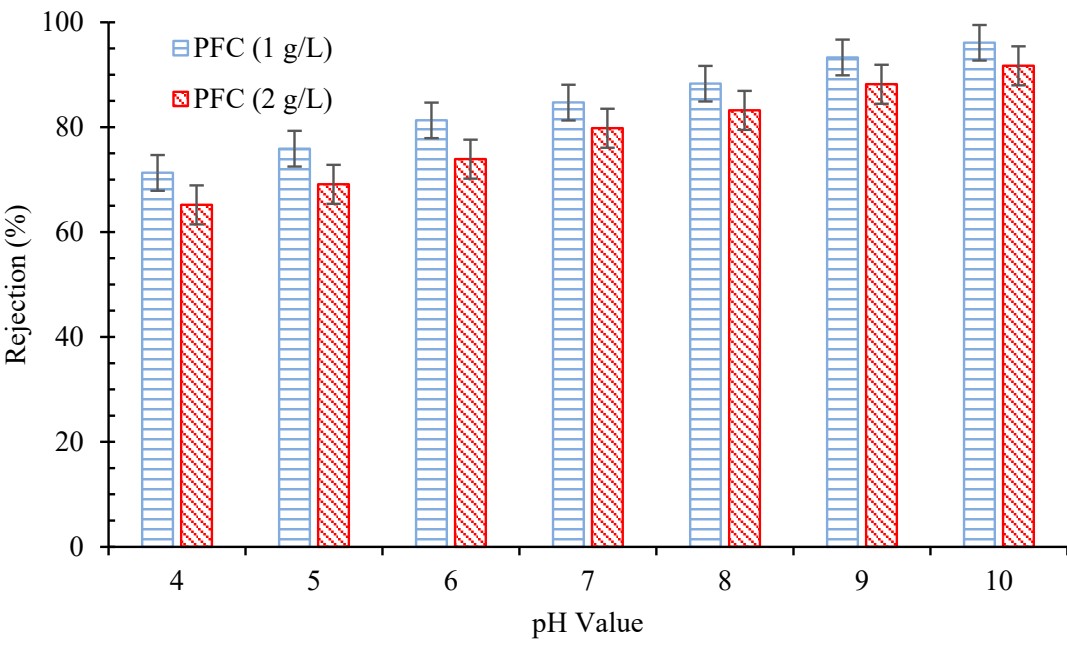

**Figure 3.** The comparison of the rejection values at different pH values using different concentrations of PFC. The upper and lower 95% confidence intervals are shown as error bars that extend above and below the top of the mean column.

Regarding Table 3, it can be concluded that the best pH for the performance of this type of coagulant is pH = 10 which the removal efficiency for 1 and 2 g/L of PFC coagulant is 96.1% and 91.7%, respectively.

Based on Figure 3, a decrease in the removal efficiency can be seen, which can be referred to as the nature of PFC in water. As can be found, determining the optimal amount of coagulant is the next important parameter in creating suitable operating conditions in the coagulation process. This is because insufficient or excessive amounts of coagulants lead to poor coagulation performance. Therefore, determining the optimal amount of coagulant in order to reduce the cost and increase the coagulation efficiency is inevitable [45].

As can be seen in Table 4, increasing the coagulant amount to a concentration of 1 g/L increases the percentage of removal, which is 98.3% for PFS coagulant and 81.3% for PFC coagulant. Nevertheless, increasing the amount of coagulant higher than 1 g/L, not only has a non-favorable effect on improving the coagulation performance and efficiency, but also led to an increase in the process price, which can be caused by the reversal of the particle charge and the re-stability of colloids and the prevention of clot formation.

**Table 4.** The effect of coagulant amounts on the COD removal and the rejection at pH = 6.

| Coagulant Dose | 0.05 g/L | | 0.1 g/L | | 0.5 g/L | | 1 g/L | | 1.5 g/L | | 2 g/L | | 3 g/L | |
|---|---|---|---|---|---|---|---|---|---|---|---|---|---|---|
| | COD | %R | COD | %R | COD | %R | COD | %R | COD | %R | COD | %R | COD | %R |
| PFS | 4230 | 35.1 | 3812 | 52.9 | 3122 | 69.2 | 2166 | 98.3 | 1542 | 68.3 | 1191 | 66.5 | 563 | 58.4 |
| PFC | 3612 | 48.3 | 3130 | 51.6 | 2855 | 59.7 | 1888 | 81.3 | 1239 | 73.9 | 953 | 73.4 | 366 | 61.1 |

Another interesting result that can be seen in Figure 4 is that using coagulant at less than 1 g/L, using PFS is more efficient than PFC; meanwhile, in more than 1 g/L of coagulant, this effect is reversed and the use of PFS will be less efficient than PFC.

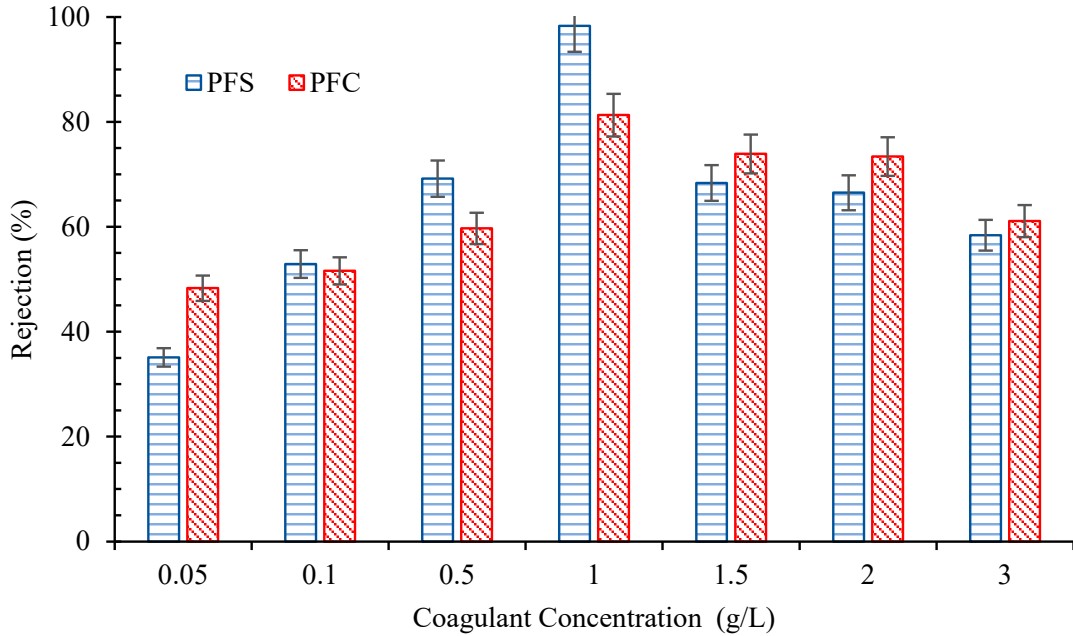

**Figure 4.** The effect of coagulant concentration on the rejection at pH = 6 using different coagulants. The upper and lower 95% confidence intervals are shown as error bars that extend above and below the top of the mean column.

The obtained results from the previous sections can be used to check the performance of coagulants, and it can be concluded that PFS has a better performance than PFC coagulant with a small difference in pH and the appropriate amount obtained for both coagulants [46].

By performing adsorption tests in different contact times up to 300 min and a time interval of 30 min, at a constant temperature of 25 °C, the COD reduction rate of the water-oil emulsion was investigated. The results are shown in Figure 5.

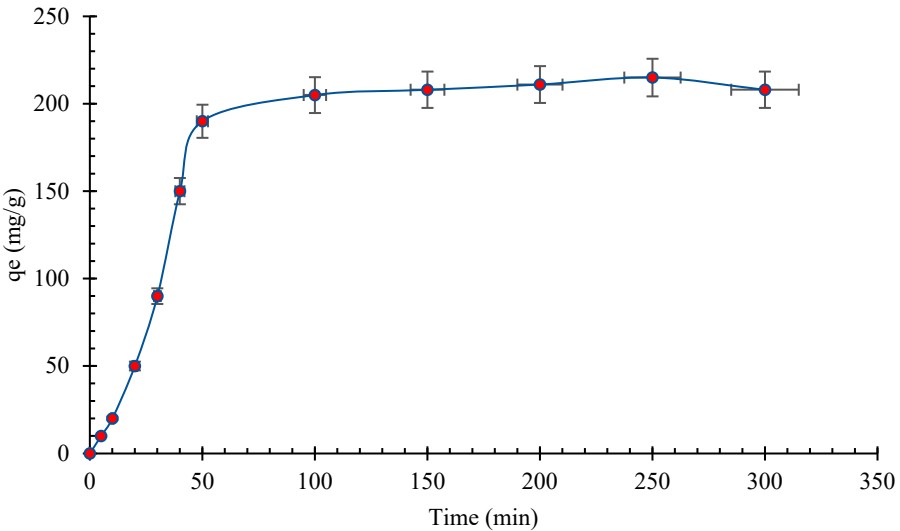

**Figure 5.** Equilibrium time in surface adsorption process at pH = 6 and T = 25 °C, pollutant concentration = 6000 ppm. The upper and lower 95% confidence intervals are shown as error bars that extend above and below the top of the mean column.

As can be seen from Figure 5, in the constant operating conditions, the time required to reach the equilibrium in the surface adsorption process is about 60 min, and this is affected by the high concentration of oil present in the sample to be removed. Therefore, the activated carbon granule adsorbent is saturated in a short time, and it is not possible to achieve a high removal efficiency with this amount of adsorbent. The adsorption capacity is calculated as Equation (2); where $V$ is the volume of the solution and $m$ is the adsorbent mass; and $C_0$ and $C_e$ are the initial and the final concentrations, respectively.

$$q = \frac{(C_0 - C_e) \times V}{m} \tag{2}$$

Figure 6 shows the effect of adsorbent concentration on the percentage of removal efficiency in surface adsorption at pH = 6 and T = 25 °C. As can be seen in Figure 6, with the increase in the amount of adsorbent, the removal efficiency and reduction of COD increased, but the desired removal rate was achieved in high amounts of activated carbon granules. For example, at a concentration of 20 g/L of the adsorbent, only 48% removal can be achieved, which may not be economical to use this amount of adsorbent due to the problem of its recovery.

According to the results, it can be concluded that, to remove the oil particles in the oil-water emulsion and reduce the COD value, a small amount of coagulant can be used. In other words, it was observed that by using a small amount of coagulant the removal efficiency of more than 90% can be achieved. This observation was made in spite the fact that only 48% of COD could be treated using the activated carbon granules as an adsorbent, which indicates the weaker performance of this method as a pre-treatment in removing the high concentration of pollutant.

As it was stated previously, the polymer grafting method has been used on the surface of nanoparticles in order to reduce their accumulation. In the first step, by adding APTMS, which acts as an activator, amine functional groups ($NH_2$) are formed on the surface of the nanoparticle ($TiO_2$-APTMS). In the continuation of the reaction between $TiO_2$-APTMS particles and ammonium persulfate, a free radical compound is formed, and as a result, methacrylic acid polymer begins to form on the surface of $TiO_2$-APTMS particles. Finally,

after polymerization, TiO$_2$-g-PMAA particles are obtained. To check the structure of nanoparticles, the Fourier-transform infrared spectroscopy (FTIR) test was applied to ensure that the reactions are carried out completely. This analysis shows well the groups created on the nanoparticle surface. Figure 7 shows the results of FTIR test for pure TiO$_2$ nanoparticles and TiO$_2$-g-PMAA. As it is obvious according to Figure 7, in the spectrum related to TiO$_2$ nanoparticles and before surface modification, a strong adsorption peak in the range of 2989 cm$^{-1}$ is observed, which is related to hydroxyl groups. In fact, the strong adsorption observed in this range indicates the presence of countless Si-O-H groups on the surface of nanoparticles.

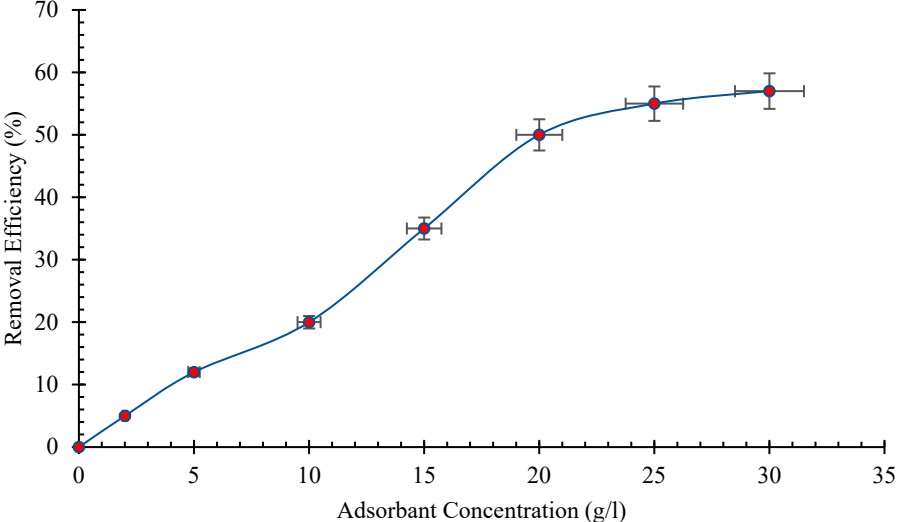

**Figure 6.** The effect of adsorbent concentration on the percentage of removal efficiency in surface adsorption at pH = 6 and T = 25 °C, pollutant concentration = 6000 ppm. The upper and lower 95% confidence intervals are shown as error bars that extend above and below the top of the mean column.

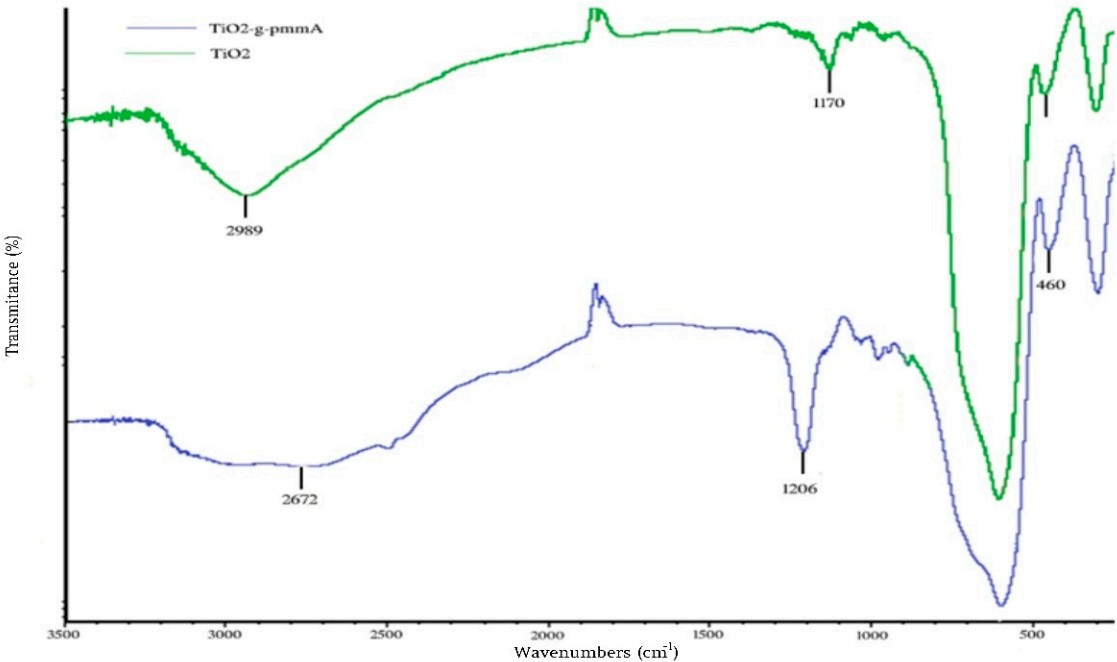

**Figure 7.** The spectra of TiO$_2$ and TiO$_2$−g−PMAA for nanoparticles with a size of 10 nm.

In the spectrum related to TiO$_2$-g-PMAA, the stretching vibrations of the carbonyl group in COOH appeared in the range of 1170 cm$^{-1}$. On the other hand, the adsorption

observed at 2672 cm$^{-1}$ indicates the hydrogen bond in the COOH group. In addition, the adsorption in the range of 1206 cm$^{-1}$ is related to the bending vibrations of the hydroxyl group in COOH. These changes in the adsorption spectrum indicate the grafting of the polymer to the nanoparticle surface and the formation of TiO$_2$-g-PMAA particles.

Figure 8 shows the effect of nanoparticle modification on the surface structure of membranes. As can be seen, the addition of nanoparticles to the PSf membrane in the polymer solution has increased the surface pores and their more regular distribution. On the other hand, by adding modified nanoparticles to the membrane structure, due to the greater compatibility that occurs in the polymer solution, more regular and better surface cavities are formed, and as a result, a membrane with more hydrophilicity and more favorable properties to has been achieved [47].

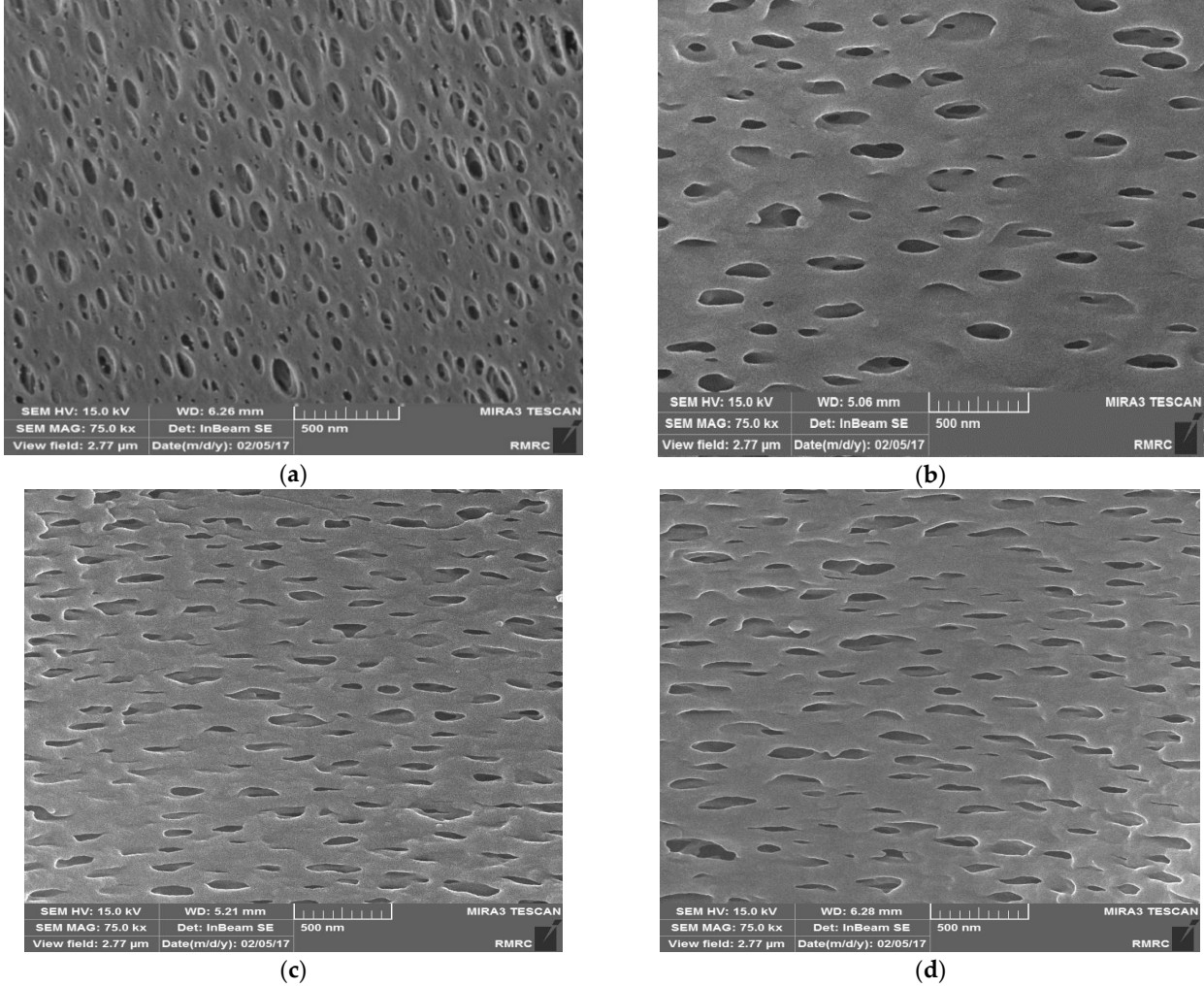

**Figure 8.** The Scanning Electron Microscope (SEM) images of the membrane surface (**a**) pure PSF; (**b**) PSF/TiO$_2$ 1%wt; (**c**) PSF/TiO$_2$-g-PMAA 1%wt; (**d**) PSF/TiO$_2$-g-PMAA 2%wt.

Figure 9 shows the SEM images of the cross-section of PSf, PSf/TiO$_2$ (1%wt), PSf/TiO$_2$-g-PMAA (1%wt), and PSf/TiO$_2$-g-PMAA (2%wt) membranes. As can be seen, the amount of porosity in the membrane increases with the addition of nanoparticles. All membranes show an asymmetric structure including a dense upper layer and a porous finger-like sub-layer. Since the pure PSf membranes have less surface porosity, they limit the penetration of larger amounts of water to the sub-layer and as a result, large porous structures are formed in the sub-layer. Nevertheless, with the addition of nanoparticles in PSf/TiO$_2$ and PSf/TiO$_2$-g-PMAA membranes, finger-like pores are found, which is caused by more surface porosity and more penetration of water into the substrates.

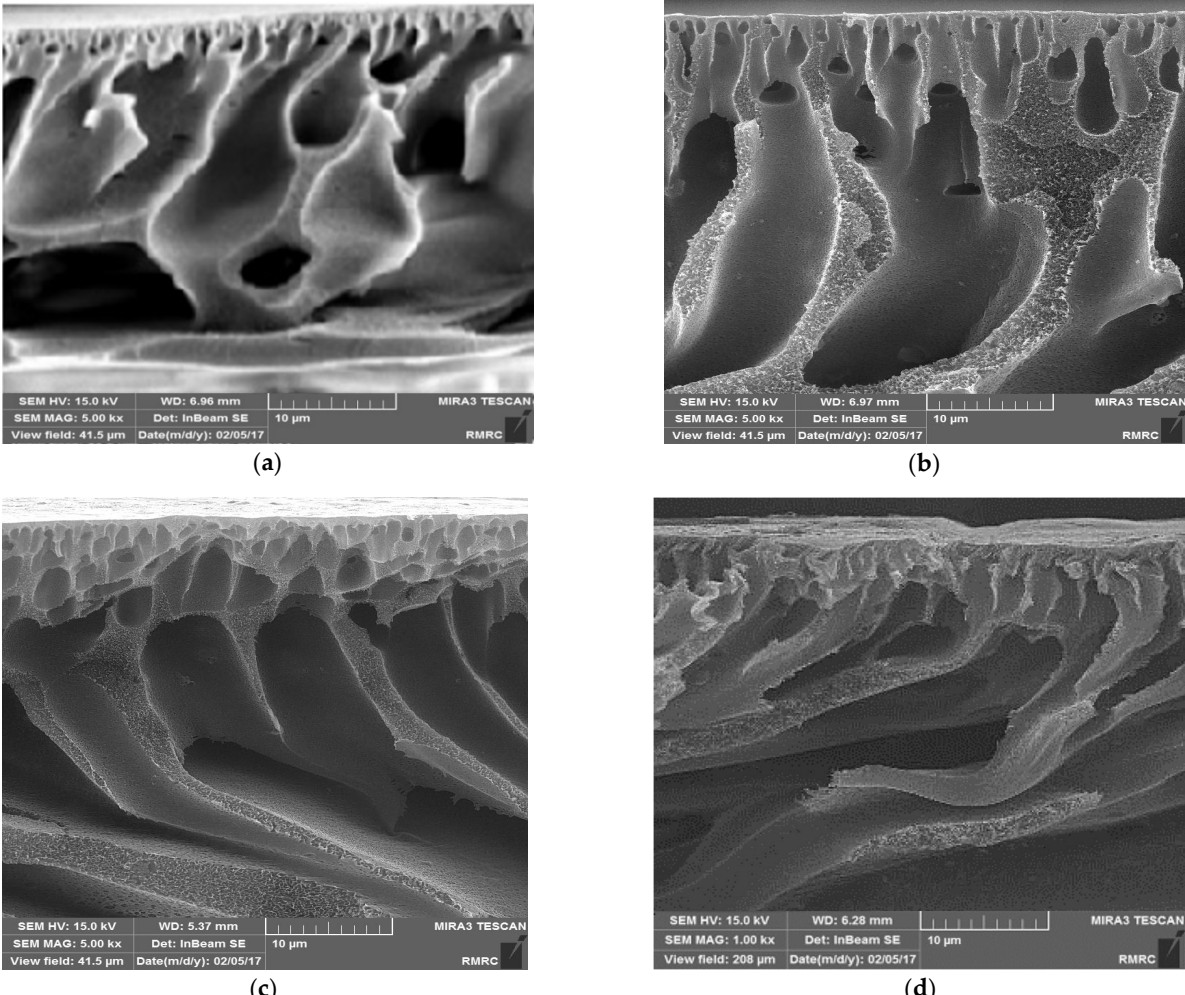

**Figure 9.** The cross-SEM images of the membrane surface (**a**) pure PSF; (**b**) PSF/TiO$_2$ 1%wt; (**c**) PSF/TiO$_2$-g-PMAA 1%wt; (**d**) PSF/TiO$_2$-g-PMAA 2%wt.

It should be noted that in the pre-treatment section, PFC coagulant with different concentrations is used, which leads to the reduction of COD of the feed. According to the pilot tests which are carried out by changing the operating parameters, the flux diagram is obtained and compared in Figure 10. As can be seen, initially, the permeate flux decreases with a relatively large slope, and then this flux drop gradually decreases. Then, after about 40 min, it reaches a constant value and becomes stable. The reason for this decrease in flux is the accumulation and settling of oil particles on the surface of the membrane [48]. Furthermore, due to changes in applied pressure and the ability to change the shape of oil droplets, there is a possibility of oil droplets penetrating into the membrane pores, which can also lead to a decrease in flux.

As can be seen from Figure 10, increasing the amount of pollutant reduces the amount of permeate flux, and also the slope of flux reduction is much higher at the beginning of the experiment. Furthermore, the results show that increasing the pressure can increase the amount of permeate flux. Therefore, decreasing the inlet pressure and increasing the amount of pollutant have a negative effect on the process. Pressure, as a driving force in the microfiltration process, is one of the most influential operating parameters that can have an undeniable role on the performance of the process due to the process mechanism in this separation operation.

Figure 11 shows the effect of operating pressure on the permeate flux of water-oil emulsion. As seen in Figure 11, with the increase in pressure, the permeate flux has increased up to 50%. In other words, the increase in driving force overcomes the existing

resistances and creates a significant increase in the flux. These observations show that generally higher pressures lead to higher fluxes, and due to the increase in applied pressure, issues such as the compression of the cake on the membrane surface, which causes a decrease in the flux at high pressures, have not occurred significantly.

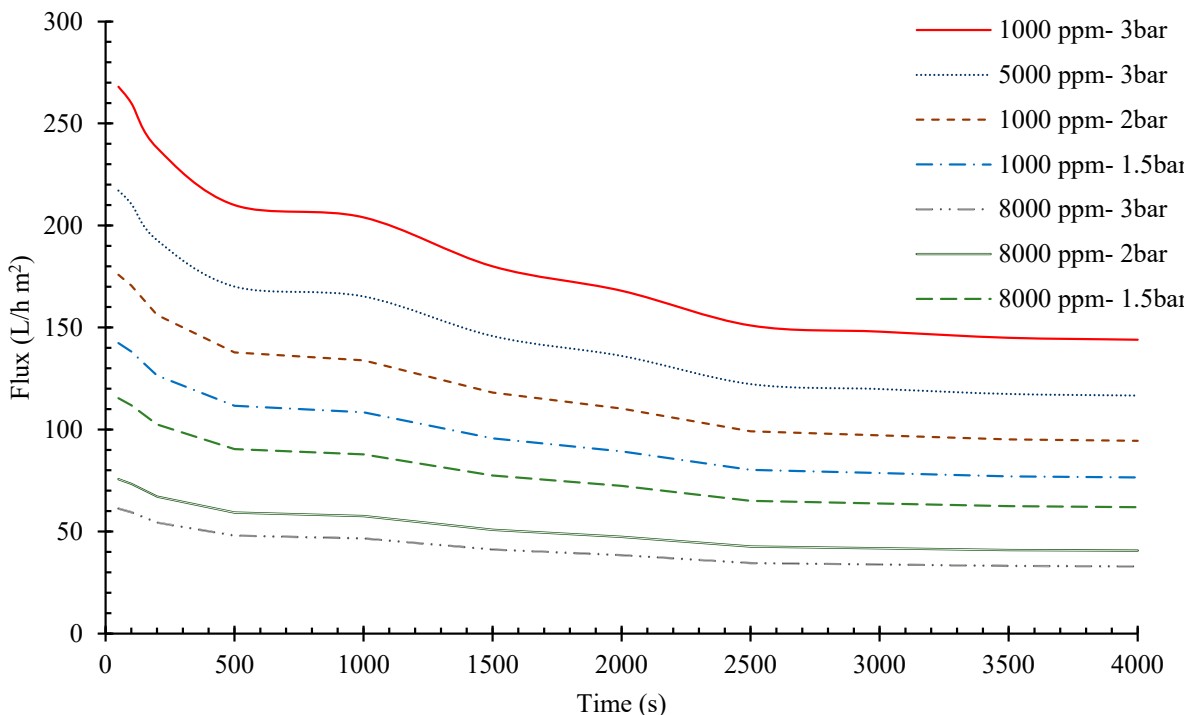

**Figure 10.** The effect of operational conditions on membrane performance.

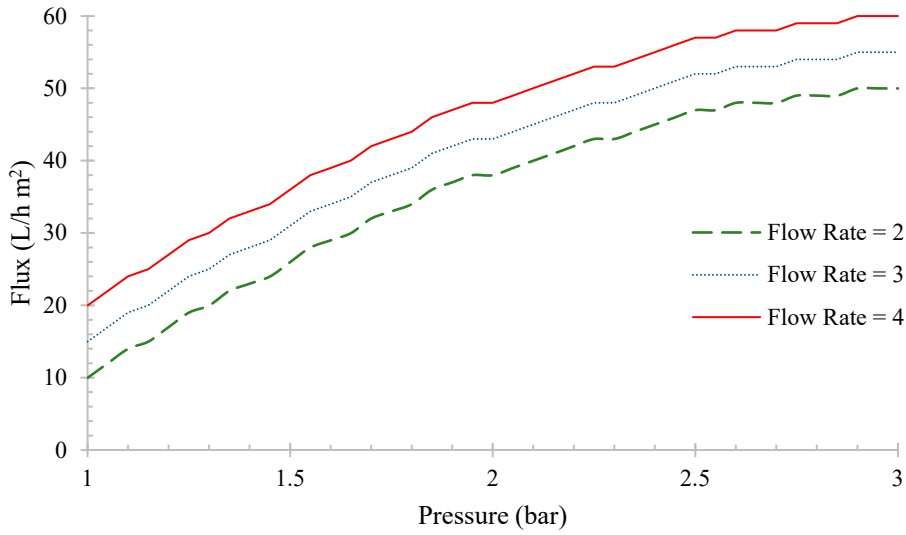

**Figure 11.** The effect of operating pressure on the flux of water-oil emulsion at constant flow rate and feed concentration.

Based on the pressure effects on the flux at a constant flow rate and feed concentration, it can be concluded that the effect of the cake layer accumulation on the membrane surface is only in the steeper slope of the flux decrease at the pressure of 2 bar compared to 1 bar. It also can be understood that by increasing the driving force, the emulsion particles were pushed towards the membrane with more force, and this caused an increase in the membrane fouling but, as can be seen, the increase in the driving force ultimately led to an increase in the flux [49].

It should be noted that above a pressure of 3 bar, increasing the pressure did not have a significant effect on increasing the flux, and the changes in the flux reached a constant level, which was also affected by the entry of oil particles into the membrane pores and the increase in the membrane fouling rate. According to economic issues and reduction of operating costs, applying less pressure to achieve the same flux can be important. Figure 12 shows the effect of increasing pressure on the permeate flux at constant feed concentration and flow rate.

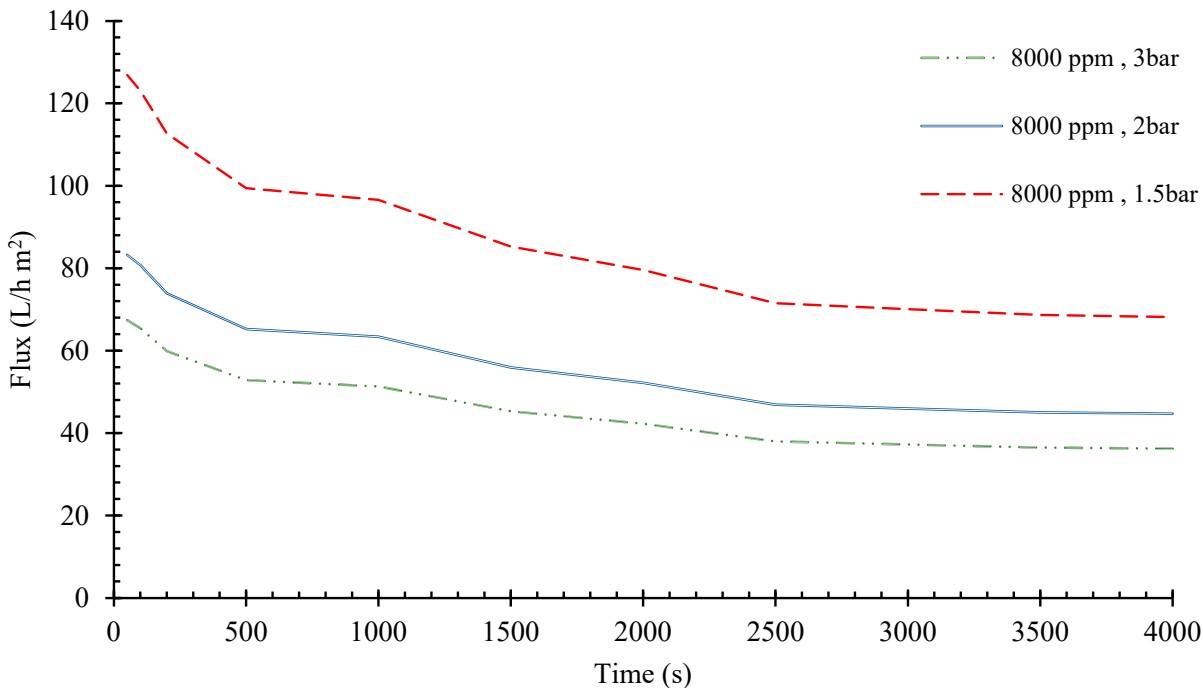

**Figure 12.** The effect of increasing pressure on the permeate flux at constant feed concentration and flow rate.

Figure 13 shows the effect of flow rate on the flux of water-oil emulsion at constant pressure and feed concentration. As can be seen from Figure 13, with the increase in flow rate, the permeate flux has increased up to 100%. In other words, the increase in flow rate overcomes the existing resistances and creates a significant increase in the flux. These observations show that generally, higher flow rate led to higher fluxes, and due to the increase in applied pressure, issues such as the compression of the cake on the membrane surface, which causes a decrease in the flux at higher flow rates, have not occurred significantly. It also can be understood that by increasing the driving force, the emulsion particles were pushed towards the membrane with more rates of the feed flow, and this caused an increase in the membrane fouling, but as can be seen, the more increase in the driving force ultimately led to an increase in the flux [50].

As can be seen in Figure 13, the highest amount of permeate flux was obtained at a flow rate of 5 L/min and a pressure of 3 bar. On the other hand, at a constant pressure of 1 bar, the increase in flow rate increased the flow rate by 67%, and at a pressure of 2 bar, by 61% [51,52].

As it can be seen in Figure 14, with an increase in COD concentration of the feed or the oil concentration in the feed emulsion, the flux decreases, which can be referred to the concentration polarization [53,54].

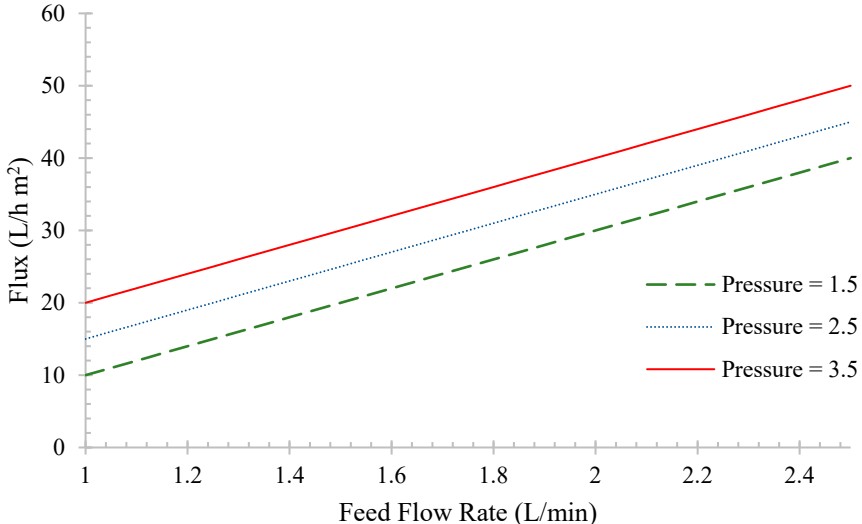

**Figure 13.** The effect of flow rate on the flux of water-oil emulsion at constant pressure and feed concentration.

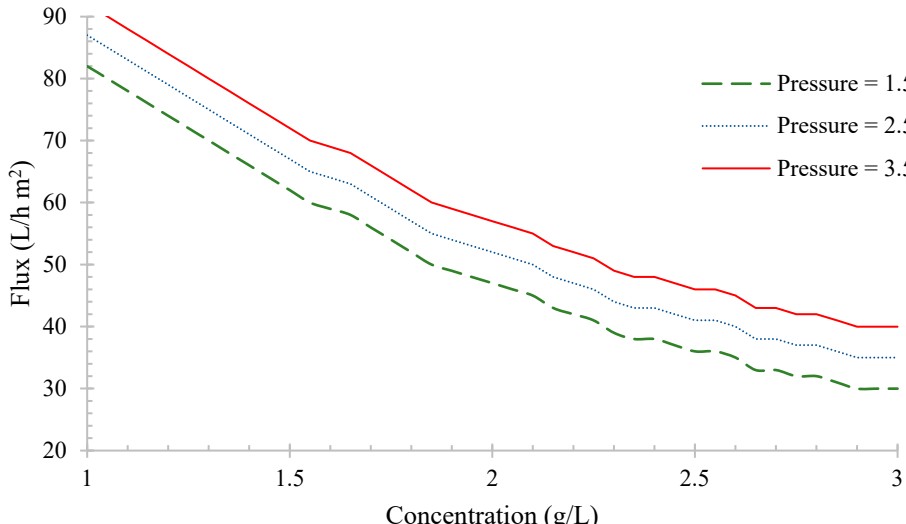

**Figure 14.** The effect of COD value of feed on the flux of water-oil emulsion at constant pressure and flow rate.

Based on the results, the effect of operational conditions on the oil rejection, the increase in flow rate has led to a decrease in the amount of repulsion, which can be concluded that the increase in flow rate is due to concentration polarization and the creation of a cake layer on the surface. The effect of operating pressure on the oil rejection results indicates that the effect of increasing the pressure of 2 bar on reducing the oil rejection by approximately 2%. However, this amount of reduction in the quality permeate flow is negligible compared to the amount of 98% removal achieved by the membrane at this amount of pressure and flow rate. Furthermore, the effect of increasing the feed concentration on membrane performance shows the improvement of the oil rejection by up to 5%, which can be caused by the effect of increasing the oil concentration on the surface of the membrane and creating a dense, thick layer, which resists against the permeate of oily particles through the membrane [55,56].

Table 5 shows the performance of pure and modified PSf membranes at optimal and constant pressures in the treatment of primary and pre-treated feed. As can be seen in Table 5, it can be stated that the modification of the membrane has greatly contributed to the increase of the permeate flux. Moreover, the pre-treatment process has led to an increase in the permeate flux in all cases [57–60].

**Table 5.** The performance of pure and modified PSf membrane at optimal and constant pressure in the treatment of primary and pre-treated feed.

| Membrane | Feed Concentration | Pressure | Flow Rate | Permeate Flux | Rejection |
|---|---|---|---|---|---|
| Pure PSf | 8000 ppm (pre-treated) | 3 bar | 4 L/h | 453 (L/h m$^2$) | 95% |
| | 8000 ppm (primary feed) | 3 bar | 4 L/h | 412 (L/h m$^2$) | 96% |
| Modifed PSf | 8000 ppm (pre-treated) | 3 bar | 4 L/h | 563 (L/h m$^2$) | 99% |
| | 8000 ppm (primary feed) | 3 bar | 4 L/h | 572 (L/h m$^2$) | 99% |

## 4. Conclusions

To describe the adsorption process, a variety of studies have been carried out. However, no extensive study has been carried out to investigate the effect of the pre-treatment method on oily wastewater. The purpose of this work was to improve the performance of the membrane process in the treatment of oily wastewater using coagulation and surface adsorption as pre-treatment. In this way, the membrane modification and the operational parameters have been investigated on the microfiltration membrane system. To investigate the coagulation process, two types of coagulants, polyferric sulfate (PFS) and polyferric chloride (PFC), were used in different amounts at different pH values. It was found that upon using coagulant of less than 1 g/L, the in the case of using PFS is more efficient than PFC; meanwhile, in more than 1 g/L of coagulant, this effect is reversed and the use of PFS will be less efficient than PFC. The adsorption studies show that with the increase in the amount of adsorbent, the removal efficiency and reduction of COD increased, but the desired removal rate was achieved in high amounts of activated carbon granules. For example, at a concentration of 20 g/L of the adsorbent, only 48% removal can be achieved, which may not be economical to use this amount of adsorbent due to the problem of its recovery. It was found that the PFS coagulant has shown a better performance as a pre-treatment method than PFC coagulant and activated carbon adsorbent.

It was also explained that due to the high hydrophilicity of the PSf membranes, it is necessary to modify the polymer using TiO$_2$ nanoparticles. For this purpose, the radical polymerization method was used to graft polymethacrylic acid to the surface of the nanoparticle and modify its surface. Finally, the membranes were fabricated using NIPS method to treatment of oily wastewater. The effect of operating conditions on membrane performance was also investigated. The change of operating conditions during the microfiltration process shows the positive effect of increasing the pressure and flow rate up to 3 bar and 5 L/min on the flux passing through the modified membrane and pre-treated effluent compared to the untreated primary effluent. However, this increase in pressure and flow rate does not have a favorable effect on oil rejection. In addition, by comparing the performance of the pure PSf membrane and the modified one by TiO$_2$ nanoparticles, it can be concluded that this also affects the performance of the membrane and the permeate flux. The optimized membrane was applied in the purification of synthetic samples using different feed concentration. In order to increase the efficiency of oil removal, it is suggested to use ion exchange resins or modified activated carbon as a pre-treatment method of high-concentration oil in water emulsion.

This work was the first step in the pre-treatment effect on the adsorption process, which can lead to optimization of the adsorption process. It is suggested that in the future, with the development of more effective coagulants, the simultaneous processes to be performed and the possibility of oily wastewater treatment using the adsorption process to be investigated.

**Author Contributions:** For Conceptualization, H.S.K.A. and S.M.; methodology, H.S.K.A. and S.M.; software, D.A. and S.Z.; validation, H.S.K.A. and S.Z.; investigation, H.S.K.A.; resources, S.M.; data curation, H.S.K.A.; writing—original draft preparation, S.Z. and M.S.; supervision, validation, and writing—review and editing, D.A. and H.M.S.; project administration, D.A. and H.M.S.; funding acquisition, H.S.K.A. All authors have read and agreed to the published version of the manuscript.

**Funding:** This research received no external funding.

**Institutional Review Board Statement:** Not applicable.

**Informed Consent Statement:** Not applicable.

**Data Availability Statement:** Not applicable.

**Conflicts of Interest:** The authors declare no conflict of interest.

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
