# Peer review of "Wastewater Management Using Coagulation and Surface Adsorption through Different Polyferrics in the Presence of TiO2-g-PMAA Particles"

_water, doi:10.3390/w15010145_

Round 1

Author Response

General Remarks Recommendation: Minor Revision

Summary: In this manuscript, the authors attempted to improve the performance of the membrane process in the treatment of oily wastewater. Specifically, the effects of pre-treatment, membrane modification, and operational parameters have been investigated on the microfiltration membrane system. The authors evaluated two coagulation and surface absorption methods using polyferric chloride (PFC) and polyferric sulfate (PFS) coagulants and activated carbon adsorbent as pre-treatment methods. Moreover, the modified TiO2-g-PMAA membrane performance was investigated. The topic well falls within the scope of Water. It is critical to developing an innovative method for the treatment of oily wastewater. Overall, the authors have conducted many experiments with good experiential designs. Only a minor revision is needed before publication. Some issues are listed below for authors to consider for the revisions.

Comment 1: In line 48, are there any references supporting that membranes can be considered as an attractive method for oily wastewater treatment due to their low cost and energy consumption?

Answer:

Thank you for your comment. Yes, there are lots of references supporting that the membranes can be considered as an attractive method for oily wastewater treatment due to their low cost and energy consumption. Some references were cited as Ref 11-13.

Comment 2: Please explain why “insufficient or excessive amounts of coagulants lead to poor coagulation process” in lines 221-222.

Answer:

Thank you for your comment. This question shows your high knowledge and your reviewing accuracy. This phenomenon was one of the interesting results of this research. If a small amount of coagulant concentration is applied in the coagulation process, its amount is not enough to remove the pollution and as a result the rejection percentage will be low. But if the concentration of coagulant is high, these materials attracted to each other because of their tendency, and do not disperse well in the solution, as a result, they cannot perform properly in the coagulation process. Therefore, insufficient or excessive amounts of coagulants lead to poor coagulation process which is explained in Line 239-244.

Comment 3: Error bars are missing in Figures 2, 3, 4, 5, and 6 of this manuscript.

Answer:

Thank you for your comment. The comment was applied and the error bars were added to the mentioned Figures.

Comment 4: Please proofread the authors’ manuscript before the final submission. There are some typos in the manuscript. For example, the “1206 cm cm-1” should be “1206 cm-1 ” in line 288.

Answer:

Thank you for your comment. The comment was applied and the whole manuscript was rewritten, modified and edited. Moreover, the manuscript was edited by a native English speaker who is a professional manuscript editor.

Comment 5: The Conclusion section of a manuscript should include a summary of the results, the potential limitation of this research, and the recommendation of future research directions for peers. However, only a summary of the results was covered in the Conclusion section of this manuscript. It would be great to add the potential limitation of this research and the recommendation of future research directions for peers.

Answer:

Thank you for your valuable advice. The conclusions section was modified and rewritten as follows:

“For the surface adsorption process, a wide range of studies have been carried out to describe the adsorption process however, no extensive study has been carried out to investigate the pre-treatment method effect on the surface absorption process. Therefore, the coagulation and surface absorption methods have not been fully understood. The purpose of this work is to improve the performance of the membrane process in the treatment of oily wastewater. For this purpose, the effects of pre-treatment, membrane modification and operational parameters have been investigated on the microfiltration membrane system. In this work, the performance of coagulation and surface adsorption as a pre-treatment of water-oil emulsion microfiltration was studied. In his way, two types of coagulants, polyferric sulfate (PFS) and polyferric chloride (PFC), were used in different amounts at different pH values. It was also explained that due to the high hydrophilicity of the PSf membranes, it is necessary to modify the polymer. One of the ways to modify these kinds of polymeric membranes is to use nanoparticles. For this purpose, the radical polymerization method was used to graft polymethacrylic acid to the surface of the nanoparticle and modify its surface. Finally, the membranes were fabricated using nonsolvent-induced phase separation (NIPS) method to separate water-oil emulsion. The effect of operating conditions on membrane performance was also investigated. The PFS coagulant has shown a better performance as a pre-treatment method than PFC coagulant and activated carbon adsorbent. Also, the change of operating conditions during the microfiltration process shows the positive effect of increasing the pressure and flow rate up to 3 bar and 5 L/min on the flux passing through the modified membrane and pre-treated effluent compared to the untreated primary effluent. However, this increase in pressure and flow rate does not have a favorable effect on the oil rejection. In addition, by comparing the performance of the pure PSF membrane and modified one by TiO2 nanoparticles, it can be concluded that this also affects the performance of the membrane and the permeate flux. The optimized membrane was applied in the purification of synthetic samples using different feed concentration. In order to increase the efficiency of oil removal, it is suggested to use ion exchange resins or the modified activated carbon as a pre-treatment method of high-concentration oil in water emulsion.

This work was the first step in the pre-treatment effect on the adsorption process, which can lead to the optimizing the adsorption process. It is suggested that in the future, with the development of more effective coagulants, the simultaneous processes to be performed and the possibility of oily wastewater treatment using the adsorption process to be investigated.”

Reviewer 2 Report

The main aim of the work was to use two coagulation and surface absorption methods using polyferric chloride (PFC) and polyferric sulfate (PFS) coagulants and activated carbon absorbent as pretreatment methods. For investigation, the modified TiO2-g-PMAA membrane was prepared and the membrane performance was studied by changing the operational parameters.

The work is interesting and the significance of the manuscript content is high. Nevertheless, before the publication, the manuscript should be improved. For this purpose, please, see the comment below:

1. The information presented in the Introduction is not sufficient. Indeed, the issue of the oily wastewater treatment and separation is not well discussed. 

The Authors wrote:

"So far, various methods have been studied and investigated in the field of oily wastewater treatment, among the common methods are flotation [2], separation based on gravity [3], surface absorption, coagulation and flocculation [4], ozonation [5], etc."

and then

"Recently, membranes have been reported as an attractive method for oily wastewater treatment, due to their low cost and energy consumption. However, these methods have disadvantages such as fouling and concentration polarization, which can reduce the lifeof the membrane. Therefore, in order to increase the membrane efficiency, it is possible to act in two ways: a) preventing the particles in the effluent from reaching the surface of the membrane b) surface washing of the membrane [7]."

This information is not complete. Recently, several papers focused on the treatment of oily wastewaters have been published and they should be discussed in the manuscript, for instance:

 - The application of MF process for separation of oily wastewater:

DOI: 10.3390/fib9060035

 - The application of UF process for separation of oily wastewater:

DOI: 10.1016/j.seppur.2020.118259

- The application of MD process for separation of oily wastewater:

DOI: 10.3390/membranes11120988

DOI: 10.3390/membranes12040351

The literature review should be better performed, which will allow to refer to many more recently published papers. This, in turn, will allow the reader to better understand the subject.

2. The aim of the work should be presented in more detail, for instance: which process the Authors studied to separate the oily wastewaters.

3. The novelty of the work should be emphasized.

4. The obtained results should be discussed in more detail. They should be compared woth those available in the literature.

5. Minor errors should be corrected, for instance:

- no subscript in: TiO2, Al2O3, ZrO2, SiO2 and Fe2O3 etc.,

- no capital letter in: Table 1. specifications of the obtained Activated carbon etc.

Author Response

The main aim of the work was to use two coagulation and surface absorption methods using polyferric chloride (PFC) and polyferric sulfate (PFS) coagulants and activated carbon absorbent as pretreatment methods. For investigation, the modified TiO2-g-PMAA membrane was prepared and the membrane performance was studied by changing the operational parameters.

The work is interesting and the significance of the manuscript content is high. Nevertheless, before the publication, the manuscript should be improved. For this purpose, please, see the comment below:

Comment 1: The information presented in the Introduction is not sufficient. Indeed, the issue of the oily wastewater treatment and separation is not well discussed. 

The Authors wrote:

"So far, various methods have been studied and investigated in the field of oily wastewater treatment, among the common methods are flotation [2], separation based on gravity [3], surface absorption, coagulation and flocculation [4], ozonation [5], etc."

and then

"Recently, membranes have been reported as an attractive method for oily wastewater treatment, due to their low cost and energy consumption. However, these methods have disadvantages such as fouling and concentration polarization, which can reduce the life of the membrane. Therefore, in order to increase the membrane efficiency, it is possible to act in two ways: a) preventing the particles in the effluent from reaching the surface of the membrane b) surface washing of the membrane [7]."

This information is not complete. Recently, several papers focused on the treatment of oily wastewaters have been published and they should be discussed in the manuscript, for instance:

 - The application of MF process for separation of oily wastewater: DOI: 10.3390/fib9060035

 -The application of UF process for separation of oily wastewater: DOI: 10.1016/j.seppur.2020.118259

- The application of MD process for separation of oily wastewater: DOI: 10.3390/membranes11120988

- DOI: 10.3390/membranes12040351

The literature review should be better performed, which will allow to refer to many more recently published papers. This, in turn, will allow the reader to better understand the subject.

 Answer:

Thank you for your comment. The comment was applied and some other references were also studied. The mentioned articles were used to enrich the literature review section and were cited. Moreover, the Introduction section was rewritten and well discussed to allow the reader to better understand the subject.

Comment 2: The aim of the work should be presented in more detail, for instance: which process the Authors studied to separate the oily wastewaters.

 Answer:

Thank you for your comment. In this manuscript, we attempted to improve the performance of the membrane process in the treatment of oily wastewater. Specifically, the effects of pre-treatment, membrane modification, and operational parameters have been investigated on the microfiltration membrane system. The authors evaluated two coagulation and surface absorption methods using polyferric chloride (PFC) and polyferric sulfate (PFS) coagulants and activated carbon adsorbent as pre-treatment methods. Moreover, the modified TiO2-g-PMAA membrane performance was investigated. It should be noted that it is critical to developing an innovative method for the treatment of oily wastewater environmentally. The gap of the previous studies and the aim of the work was presented as Line 104-116.

Comment 3: The novelty of the work should be emphasized.

 Answer:

Thank you for your comment. The comment was applied and the novelty of the work was emphasized in the last paragraph of the introduction section, abstract and the conclusion. It should be noted that for the surface adsorption process, a wide range of studies have been carried out to describe the adsorption process however, no extensive study has been carried out to investigate the pre-treatment method effect on the surface absorption process. Therefore, the coagulation and surface absorption methods have not been fully understood. In this paper, in order to study the surface adsorption process, different pre-treatment methods have been performed for the removal of oily pollutants. In this regard, the obtained experimental data have been carefully discussed to suggest the optimum method as Line 104-116.

Comment 4: The obtained results should be discussed in more detail. They should be compared with those available in the literature.

  Answer:

Thank you for your comment. The comment was applied and results were discussed completely and were compared with other researches which is highlighted in red color

Comment 5: Minor errors should be corrected, for instance:

- no subscript in: TiO2, Al2O3, ZrO2, SiO2 and Fe2O3 etc.,

- no capital letter in: Table 1. specifications of the obtained Activated carbon etc.

  Answer:

Thank you for your comment. Your comment was applied and whole the manuscript was edited and modified properly.

Round 2

Reviewer 2 Report

The manuscript has been significantly improved. Therefore, I recommend it for publication in current form.

Author Response

 Thank you so much for your positive feedback. We truly appreciate your effort for taking the time to review our manuscript.